# Effect of Pomegranate Extract Consumption on Satiety Parameters in Healthy Volunteers: A Preliminary Randomized Study

**DOI:** 10.3390/foods11172639

**Published:** 2022-08-31

**Authors:** Angela Stockton, Emad A. S. Al-Dujaili

**Affiliations:** 1Dietetics, Nutrition and Biological Sciences, Queen Margaret University, Edinburgh EH16 4TJ, UK; 2Centre for Cardiovascular Science, Queen’s Medical Research Institute, University of Edinburgh, Edinburgh EH16 4TJ, UK

**Keywords:** pomegranate extract, polyphenols, obesity, satiety, appetite, visual analogue scale

## Abstract

There has been an increasing interest in nutraceuticals and functional foods in reducing appetite and to lose weight. We assessed the effect of oral pomegranate extract (PE) and PE juice (PJ) intake vs. placebo on satiety parameters in healthy volunteers. Twenty-eight subjects (mean age 34.5 ± 13.7 years, body mass index [BMI] 25.05 ± 3.91 kg/m^2^) were randomized to 3-week priming supplementation with PE (Pomanox^®^) or placebo. On week 3, satiety parameters were determined on 1 testing day after participants ingested a breakfast and a lunch meal with PJ juice, using 100-mm visual acuity scales (VAS) for hunger, desire to eat, fullness and satisfaction. Meal quality and palatability were also tested. The desire to eat was less at all time points in the PJ juice with PE priming group and participants were also less hungry (*p* = 0.044) than those who consumed placebo. There was an overall significant difference between the groups (*p* < 0.001). Participants in the PJ juice with PE priming group experienced significantly greater satisfaction (*p* = 0.036) and feeling of fullness (*p* = 0.02) than those in the placebo group. These findings suggest that consumption of PE could have the potential to modulate satiety indicators.

## 1. Introduction

The global impact of obesity on morbidity, mortality and healthcare costs is enormous, with both lifestyle and dietary habits as the key determinants in the prevalence of obesity [1]. The continuous rise in the obesity epidemic worldwide [2], coupled with the lack of success of current approaches to weight management and the uncertainty around potential drug side effects, have led to hopeful consumer interest in nutraceuticals and functional foods [3]. Yeoman and Chambers [4] suggested that “the worldwide increase in the prevalence of obesity and overweight has resulted in an urgent need to better understand the nature of satiety”. Appetite control has been highlighted as one of the most important factors involved in the success of dietary treatments of obesity. It is known that appetite sensations play an essential role in the control of energy intake [5], and, therefore, in the control of body weight [6,7].

Satiety occurs at the end of a meal and prevents further eating. Satiation takes place during a meal and determines the duration and size of the meal as well as how quickly it is consumed [8]. Appetite sensations, including hunger, fullness, prospective consumption, and desire to eat, are the appetite-related dimensions most consistently used in research [3,9,10,11]. De Graaf et al. [11] concluded from their study into biomarkers of satiation and satiety that the next important challenge in the field is to identify food ingredients which may have an impact on either satiety or satiation, with the hope that such foods may assist consumers in maintaining a healthy body weight. The capacity of specific foods and food characteristics to affect appetite (i.e., feelings of hunger and satiety) continue to be a focus of interest [12], particularly in relation to food consumption and weight management [13]. A variety of neuropeptides such as cholecystokinin and ghrelin, plus hormone secretions such as leptin, mediate these processes and influence food intake and therefore satiety [14,15].

Polyphenols have received special attention as potential complementary approaches in the management of obesity [16]. In a recent in-depth review of managing obesity through natural polyphenols [17], appetite suppressing effects of natural polyphenolic compounds have been attributed to the interplay of different mechanisms, including slowing-down secretion of appetite-stimulating hormones, inactivation of appetite sensors, modulation of melanin-concentration hormone receptors, inhibition of ghrelin secretion, increase of adiponectin levels, reduction of glucagon-like peptide 1, increase of serotonin, and modulating adipohormones and the expression of gut peptides, all involved in transferring satiety signals to the brain [18].

The consumption of pomegranate juice has grown tremendously due to its reported health benefits, which are presumably due to its high content of antioxidant polyphenols especially tannins, anthocyanins and ellagic acid derivatives [19,20]. Animal and in vitro studies have shown that pomegranate may reduce inflammation [21], lipid peroxidation, oxidative stress, blood pressure [22,23,24] and insulin resistance [25]. Pomegranate may therefore have a protective role against atherosclerosis, hypertension, cancer, diabetes type II, and obesity [24,25]. It has also been shown that the intake of pomegranate juice [26,27] and pomegranate extract reduced blood pressure, insulin resistance and stress hormones levels [28]. Pomegranate juice and pomegranate extract intake have also been reported to decrease food consumption and reduce body weight in animals [29,30]; a human study by González-Ortiz et al. [31] showed that the administration of 120 mL pomegranate juice daily for one month significantly decreased fat mass in adults. It has been suggested that one of the mechanisms by which pomegranate can participate in the management of obesity is through regulating appetite [24]. However, research investigating this association remains relatively limited. The aim of this study was to assess the effect of pomegranate extract and juice intake on satiety parameters in healthy volunteers. It was hypothesized that consumption of a pomegranate extract may reduce appetite, which could be evaluated by means of satiety parameters.

## 2. Materials and Methods

### 2.1. Study Design

This small-scale exploratory study on satiety was conducted over 1 day (morning and lunch time) as a randomized placebo-controlled trial, and in the framework of a previous randomized, double-blind controlled trial that investigated the effect of consumption of a pomegranate extract (PE) on blood pressure, insulin resistance, stress hormones and quality of life [28]. The protocol of the present satiety study was approved by the Institutional Review Board of Queen Margaret University, Edinburgh, UK. Written informed consent was obtained from all participants. The overall trial was registered in the ClinicalTrials.gov (accessed on 17 February 2019) (NCT02005939).

### 2.2. Participants

Volunteers were recruited through the university’s email research digest and by word-of-mouth recommendations. The study was open to both men and women from 18–65 years of age, with a body mass index (BMI) between 18 and 34.9 kg/m^2^ and no history of diabetes or cardiovascular diseases. Exclusion criteria were systemic disease, including heart disease, diabetes, liver or kidney dysfunction and immunological disorders; heavy smoking; abuse of drugs or alcohol; pregnancy and breastfeeding; allergy to either pomegranate (PE) or test foods; and history of weight control management within the last 2 months. Also, as recommended by Smeets et al. [32] restrained eaters recording a score of >7 in the three-factor eating questionnaires were ineligible as these subjects may control their intake of food and would therefore be unlikely to eat freely.

### 2.3. Pomegranate Extract and Placebo Capsules

The study consisted of a daily oral intake of PE capsules or placebo for 3 weeks and 1 testing day (satiety session) in the kitchen/food laboratory on week 3. Participants were randomized to the PE or the placebo group by an independent technical staff using a research randomizer website [33]. Each PE capsule contained a standardized extract of whole pomegranate fruit obtained by an eco-friendly process (Pomanox^®^ P30, Euromed S.A., Barcelona, Spain), with the following composition: 210 mg punicalagin (the recommended daily intake to provide the beneficial effects of these antioxidants) and 328 mg total polyphenols, including other pomegranate polyphenols (e.g., flavonoids, ellagic acid and 0.37 mg anthocyanins). Placebo capsules contained maltodextrin, and both PE and placebo capsules were identical in appearance, weighing 1.08 g and providing 6.52 kcal per capsule.

### 2.4. Satiety Session: Breakfast and Lunch Meals

Participants came to the kitchen/food laboratory between 8:30 and 9:30 a.m. in at least 8-h fasting conditions (water was allowed). They had previously received instructions by e-mail being advised to follow their usual diets with no restrictions but requested to refrain from alcohol or extra physical activity on the previous day. On the testing day, they received a breakfast and a lunch meal. The breakfast meal (432 kcal) consisted of 60 g of Kellogg’s Crunchy Nut Cornflakes cereal (243 kcal), 150 mL of semi-skimmed milk (69 kcal) and 150 mL of a pomegranate juice (PJ) or a placebo juice drink. The PJ (PomeGreat^®^ Pure supplied by RJA Foods, UK) contained 126 mg total polyphenols as gallic acid equivalent and 72 kcal as carbohydrates in 150 mL (sugars 12 g per 100 mL). The placebo juice was composed of diluted orange juice with very low polyphenols containing the same amount of energy as the PJ by adding sucrose. The drinks were equivalent for colour and energy (Table 1) and were served in black containers to disguise a slightly different appearance and viscosity as the PJ was thicker and slightly cloudy.

The lunch meal consisted of 1 kg of pasta with sauce (served hot) and another 150 mL of PJ to be consumed 30 min before lunch. The total energy of the lunch provided was 1057 kcal. The lunch composition is shown in Table 1. The pasta was comprised of 500 g cooked weight of pasta [Morrison’s Italian Pasta Twists (100% durum wheat semolina)], and the sauce contained 500 g of a pre-prepared pasta sauce (Tesco Bolognese Sauce, a vegetarian tomato sauce with garlic and herbs). The choice of this food quantity was based on a feasibility trial carried out in four participants, who were provided with the test meal to eat. The pre-weighed lunch portion was served after completion of the VAS at 30 min after the preload PJ as shown in Table 2 [4]. A maximum of 1 kg of pasta and sauce could be consumed [33]. A smaller food serving size was not adopted because the cue of an empty dish could prompt meal termination [34]. The homogeneity of the test meal facilitated assessment of macronutrient and energy intake [35].

### 2.5. Study Procedures

The satiety study intervention protocol is summarized in Table 2. Briefly, between 8:30 a.m. and 1:30 p.m. subjects stayed in the laboratory in controlled conditions where the breakfast and a lunch meal were served, and visual analogue scales (VAS) were recorded to assess general personal feelings for appetite and to examine the acceptability and palatability of both the juice and the test meal qualities. A 100-mm VAS with words anchored at each end expressing the most positive and negative ratings regarding satiety-related characteristics (0 = not at all, 100 = extremely) was used. Questions for general personal feelings of appetite were the following: “How hungry do you feel?”, “How strongly do you feel a desire to eat?”, “How satisfied do you feel?” and “How full do you feel right now?”. The characteristics of the quality of meal and juice were also measured using VAS and included the following five questions: “How visually appearing is the meal/juice?”, “How much do you like the smell of the meal/juice?”, “How much do you like the taste of the meal/juice?”, “How much do you like the texture of the meal/juice?” and “How palatable is the meal/juice overall?”.

Participants were kept in this controlled kitchen/food laboratory environment while completing all VAS scores. As shown in Table 2, participants recorded their VAS scores at baseline (VAS 1, 0 min) from the start of the intervention in the absence of food or juice, and then at 15-min intervals for 120 min (VAS 2–9). Consumption of the allocated preload juice took place immediately after baseline VAS was recorded. Changes from baseline were calculated by subtracting the baseline score (at 0 min) from the score at a certain time point (starting at 15 min and continuing through VAS time points until 120 min) as suggested by Veldhorst et al. [36]. A composite score of the VAS responses was used to report the overall subjective satiety scores for each treatment. Also, total meal quality was a composite score derived from the five quality-related variables (visual appeal, liked smell, like taste, like texture, and perceived overall palatability).

### 2.6. Food and Drink Diaries

A food and drink diary were used to account for dietary habits. The 3-day diet diaries were completed as previously described [23] on two occasions, once before the study began, and then on each day before, during, and after the lunch-day intervention. To monitor whether increased energy before a meal result in reduced intake at that meal, the participants’ 3-day diet diary encompassing the study day was compared to the pre-intervention consecutive 3-day diet diary [37]. Energy intake (kcal) and macronutrient intakes (i.e., protein, fat, and carbohydrates) were expressed in g and calculated from diet diary data input using the method of Astbury et al. [38].

### 2.7. Statistical Analysis

Data were analyzed using SPSS for Windows^®^ version 21.0 (Microsoft, IBM Corp., Armonk, NY, USA, 2012). Continuous normally distributed data are expressed as mean ± SD. Normality of all data was examined using the Shapiro-Wilk test. Differences in baseline characteristics between groups were explored using independent t-tests or the Mann-Whitney *U* test when the data were non-parametric. Data are generally presented as mean ± standard deviation (SD) unless otherwise stated. VAS for subjective ratings of hunger, desire to eat, satisfaction, and fullness during the 120 min postprandial period were analyzed using a mixed two-way model analysis of variance (ANOVA), with time (15, 30, 45, 60, 75, 90, 105, 120 min) as the within-subject factor, and treatment type (PE/placebo) as the between-subject factor. Food intake at the test meal was calculated from the amount of food (g) eaten at the ad libitum meal. It was hypothesized that PE priming would reduce the amount of food consumption at meal. Energy and macronutrient intakes were analyzed using NetWISP 4 (Tinuviel Software 2006, 7th Edition to the UK Composition of Foods, Anglesey, UK) [39]. Differences in energy and macronutrients were assessed using paired t-tests. Significant changes were set at *p* ≤ 0.05.

## 3. Results

### 3.1. Study Population

Twenty-nine volunteers were recruited, but one participant did not attend the satiety session for personal reasons. The study population included 28 subjects, 14 in the PE group and 14 in the placebo group. There were no significant differences in age (*p* = 0.51) and body mass index (BMI) (*p* = 0.72) between the two groups. No side effects of PE or adverse events on the screening assessment were reported by all participants. The baseline characteristics are shown in Table 3.

### 3.2. Satiety-Related Variables

VAS scores showed that there were lower levels of hunger (Figure 1) and a desire to eat (Figure 2), as well as higher levels satisfaction (Figure 3) and fullness (Figure 4), in the PE capsule group drinking PJ, compared to those who had placebo priming. At 30–90 min, participants were less hungry after consuming PE priming for 3 weeks and PJ on the test day than those who consumed placebo capsules with PJ. The desire to eat was less at all time points in participants who had taken PE capsules with PJ before lunch. These results suggest that the PE group participants with the PJ preload were generally more satisfied than the placebo group with the PJ preload, specifically 30 min after consuming lunch. Participants in the PE group consuming PJ felt less full before lunch than those in the placebo group. However, the PE group experienced more fullness than those in the PL group as lunch progressed from 30 to 120 min.

### 3.3. Meal Quality

At the 35-min period, the five variables of meal quality were recorded, with VAS scores greater than 60 in the responses to each individual question (Figure 5). There was a statistically significant difference in the overall VAS score for meal quality between the study groups, with higher scores in subjects treated with PE priming (mean 72.9 ± 12.7) than in those treated with placebo (63.8 ± 5.5) (*p* = 0.001).

### 3.4. Meal Palatability

VAS scores taken over the entire lunch period of 120 min for meal palatability showed a significance between-group difference, with higher ratings in the PE priming group with PJ preload (mean 24.0 ± 8.9 mm) than in the placebo priming group (15.9 ± 12.1) (*p* < 0.05) (Figure 6).

### 3.5. Meal Food Consumption

There was a significant difference in the mean weight of food consumed between the PE priming group (471 ± 100 g, range 447–636) and the placebo priming group (574 ± 169 g, range 569–1000) (*p* < 0.05), with lower amounts in the PE group.

### 3.6. Food and Drink Diaries

Table 4 shows the comparison of data from the 3-day diet diaries after starting PE or placebo priming and the 3-day diet diaries encompassing the satiety session day. In both study groups, there were increases in the consumption of carbohydrates, sugars, and starch, with significant differences as compared with baseline data. Increases were of a higher magnitude in the PE priming group for carbohydrates and sugars than in the placebo group, whereas changes in starch were greater for the placebo group. These increases, however, did not affect the total energy intake in either the PE group or the placebo group, with a mean difference from baseline of 196 ± 389 kcal (*p* = 0.07) and 297 ± 629 kcal (*p* = 0.12) for the PE and placebo groups, respectively.

## 4. Discussion

The purpose of this study was to conduct a preliminary testing of the potential role of PE (Pomanox^®^) in promoting satiety and reducing food intake. The results suggest that subjects in the PE group with the PJ preload were generally more satisfied than those treated with placebo. Participants were less hungry after PE capsules intake with PJ during the meal than those who consumed placebo capsules. Results of paired *t*-tests for VAS appetite measurements showed some significance between-group differences and a definite trend towards lower levels of hunger and a desire to eat, as well as higher levels of fullness and satisfaction, thus greater levels of satiety, in participants consuming PE with PJ, compared to placebo. However, the fact that these trends were noted only in the group consuming PE and juice warrants further investigation on a larger cohort.

### 4.1. Methodology and Compliance

The assessment of meal and juice palatability was carried out to eliminate any potential change in food and juice intake resulting from low scores in either the meal or juice palatability by the two groups. The mean scores for both groups were sufficiently high to confirm that the participants found the meal and the PE and PJ to be palatable. However, the meal quality was rated significantly more highly (*p* = 0.03) by participants who had Pomanox^®^ PE priming and drank the PJ than those taking PL (24.0 ± 8.9 mm versus 15.9 ± 12.1 mm, respectively). VAS tests have previously been viewed critically, and in some well-controlled studies, changes in VAS failed to predict subsequent reductions in food intake [3,9,38,40]. Another criticism of VAS is the reluctance of subjects to make full use of the scale, preferring either to avoid extreme responses or to record only their responses, thereby affecting the results [41]. Other problems associated with the use of standard VAS measurements that have been discussed by Sadoul et al. [13] included non-linearity, the inherently abstract nature of overall ‘hunger’ feelings, and the lack of quantitative interpretation of when or how much the volunteer might wish to eat. Single-item meals can also be monotonous, and thus likely to limit consumption within the study, irrespective of the enhanced satiating potential of one of the preloads. Dietary monotony may curb excessive intake but is unlikely to be commercially viable as a solution to weight management [3].

### 4.2. Satiety Mechanisms

The mechanisms underlying the potential effect of PE extract on satiety remain unclear. It has been suggested that the role of polyphenols contained in pomegranate in suppressing appetite can be similar to that of sibutramine [29]. In a study by Lei et al. [30], high-dose (800 mg/kg) pomegranate leaf extract (PLE) had an inhibitory effect on the energy intake of mice and was associated with a marked decrease in serum glucose, triglycerides (TG) and total cholesterol (TC) concentrations, along with a significant decrease in the TC/high-density lipoprotein cholesterol (HDL-C) ratio, compared to the mice fed a high-fat diet. The PLE mice also showed a significant decrease in body weight and energy intake compared to the high-fat diet control group. Sibutramine, the anti-obesity drug, was used in another arm of the same study, and produced significant effects, not only in reducing body weight and adiposity, but also in lowering circulating TG and glucose concentrations, as well as the TC/HDL-C ratio. The researchers suggested that, since the effect of PLE on energy intake was like that of sibutramine, suppressing energy intake and inhibiting the intestinal absorption of dietary fat via inhibition of pancreatic lipase activity might be potential mechanisms for the anti-obesity effect of the PE extract. We have shown previously that cacao bean extract exhibited an inhibitory effect on pancreatic lipase in vitro [42]. It could also be possible that the intense flavor generated by polyphenols in the juice [43] is responsible for inducing satiety. Therefore, designing a study using VAS to measure the effect of PE extract and PJ on sensory feelings may be useful.

### 4.3. Food Consumption

In the present study, PE priming was associated with a significantly lower amount of food consumption during the satiety session as compared with placebo priming. Although small-scale studies are not powered to detect minimal clinically important differences, and do not provide results that are clearly statistically significant [43], our results are encouraging, and provide a good foundation for any future investigations testing the direct effect of PE on satiety and amount of food intake per meal. The analysis of food and alcohol diaries showed an increase in carbohydrate and sugar intake in both groups compared to pre-intervention levels. This could be explained by the additional energy that is provided following the ingestion of a meal and juice (high in carbohydrates and sugars), which may have affected the mean intake. In fact, Rolls et al. [34] showed that people who were offered a large portion of food ate more than those offered a small portion. Whether a different preload time, inter-meal interval, higher dose of PE priming, or different meals could have produced more significant eating effects are some aspects that can be considered in future studies. Because satiation depends on both gastric and intestinal nutrient stimulation and interactions, Zaremba et al. [44] noted that an optimal dose and timing between preload and an ad libitum test meal would likely be crucial in detecting any eating-inhibitory effects.

### 4.4. Limitations and Future Directions

In future studies, a crossover trial design, where VAS could be repeated during the test meal with the alternative PE/placebo treatment, might provide more accurate results [45]. For assessing inter-individual differences in eating patterns and completing the VAS, a ‘within subject’ study design, rather than separate groups of control and test subjects, is more commonly used, with participants acting as their own controls [15]. The effect of the PJ may also have been confounded by the priming capsules consumed at the same time. This deserves further investigation. However, effectively blinding participants to the type of extract and juice consumed was beneficial for avoiding any potentially significant effects resulting from expectation bias. Future researchers should carefully consider the randomization process for more complex studies.

Compensatory eating and responses to sensory quality vary across individuals, depending on their body size, physical activity, age, and level of restraint [4]. Our study primarily recruited participants with normal weight and unrestrained eating habits. Whether the results would have been the same with an overweight/obese participant group remains to be explored. Due to the length of lunch in our study, participants had access to 500 mL of water, and should have been able to clearly identify whether they were hungry, rather than thirsty. Conversely, this may have led to erroneous feelings of satiety (due to the water consumed, rather than the food). However, the volume of water consumed should have been recorded, and considered in the interpretation of the results. The Cochrane Collaboration Systematic Review of studies concerning the dietary supplement Chitosan and weight loss [46] recommended that investigations involving BMI and appetite should have a medium-to-long term duration (8 weeks was the average period for the trials accepted for their review), and this should be considered in future studies.

Benelam [15] cautioned that, despite satiation and satiety processes having the potential to control energy intake, many individuals habitually override the signals their bodies generate. He argued that to help people control their energy intake, thus preventing weight gain, knowledge about foods, ingredients, and the dietary patterns that can enhance satiation and satiety is still potentially useful in controlling body weight and avoiding shifts towards obesity. Our study was exploratory, and not powered to detect any statistically significant effects on satiety measures, making it more difficult to identify any meaningful effects. Livingstone et al. [41] have noted that the measurement of satiety is complex, since many physical factors, such as body weight, age, gender, and menstrual cycles, can all influence appetite and energy intake [11]. Eating behavior can also be affected by other factors, such as diet or the influence of other people who may be present. Examples include the completion of a food diary, which is prone to bias, particularly towards under-reporting of energy intake and differential misreporting of the macronutrients [47]. In the current study, food diary analysis was completed using the NetWISP software programme, which required a high degree of data processing. Some input errors might have occurred, particularly those involving inaccurate nutritional substitutions of food categories that were not available within the programme [48].

### 4.5. Critical Issues

Blundell [8] emphasized that care is also required when considering the legislative framework around functional food product claims. Examples of this have been ‘freedom from hunger’ and ‘feel fuller for longer’. The European Food Safety Authority states that claims must be substantiated by scientific evidence and should be clearly understood by consumers. Since ingested foods influence appetite by a system of physiological satiety signals, functional foods could act by increasing the potency and/or duration of these signals. PE extract and PJ might be useful for public health, are readily available commercially, and appear to have no side effects. PE products could also provide a prevention and therapy alternative to pharmaceutical drugs that produce side effects or are otherwise poorly tolerated by some people. One example of this is orlistat, a drug used for weight loss, which is a highly potent pharmaceutical, and has well-recognized side effects such as diarrhoea and fatty stools. Orlistat can also interfere with the absorption of vitamins A, D, E, K and beta-carotene [49]. Moyad [50] has stated that, while companies should be given recognition for their research efforts into the effects of PE, more attention should be paid to the efficacy of experimental designs using suitable control groups.

## 5. Conclusions

Although this study indicates that PE and PJ could have potential in modulating satiety indicators, limitations of the study design, the small number of participants, and the outcomes obtained do not support drawing strong conclusions. However, the results obtained seem to be promising. Further trials with a larger number of volunteers and assessment of the effect of the PE extract and juice on satiety parameters are required.

## Figures and Tables

**Figure 1 foods-11-02639-f001:**
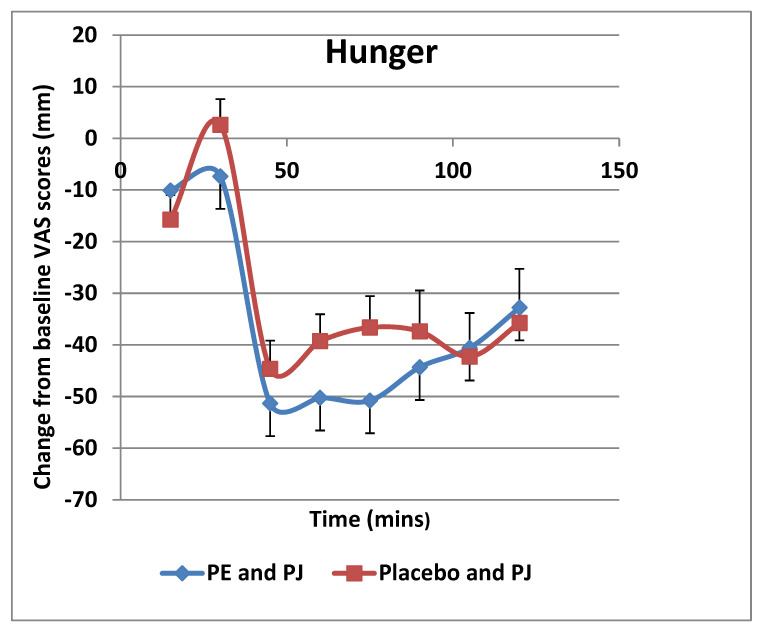
Change in hunger from baseline (mm VAS) over 120-min lunch meal period after 3 weeks of PE or placebo capsule consumption with pre-meal PJ intake. Results are expressed as mean (±SD). PE: pomegranate extract; PJ: pomegranate juice. There was an overall statistically significant difference between the two groups (*p* = 0.044) where the hunger feeling was significantly lower in the PE extract group.

**Figure 2 foods-11-02639-f002:**
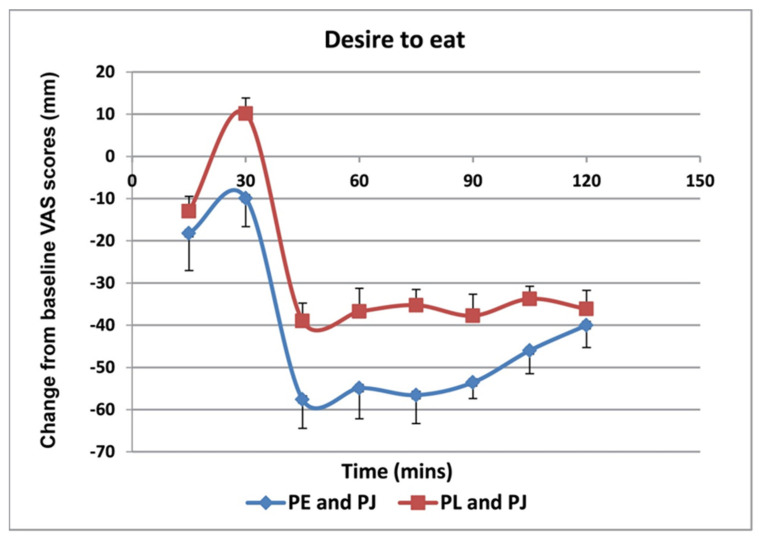
Change in hunger from baseline (mm VAS) over 120-min lunch meal period after 3 weeks of PE or placebo capsule consumption with pre-meal PJ intake. Results are expressed as mean (±SD). PE: pomegranate extract; PJ: pomegranate juice; PL: placebo capsules. There was an overall statistically significant difference between the two groups (*p* < 0.001) where the desire to eat was significantly lower in the PE extract group.

**Figure 3 foods-11-02639-f003:**
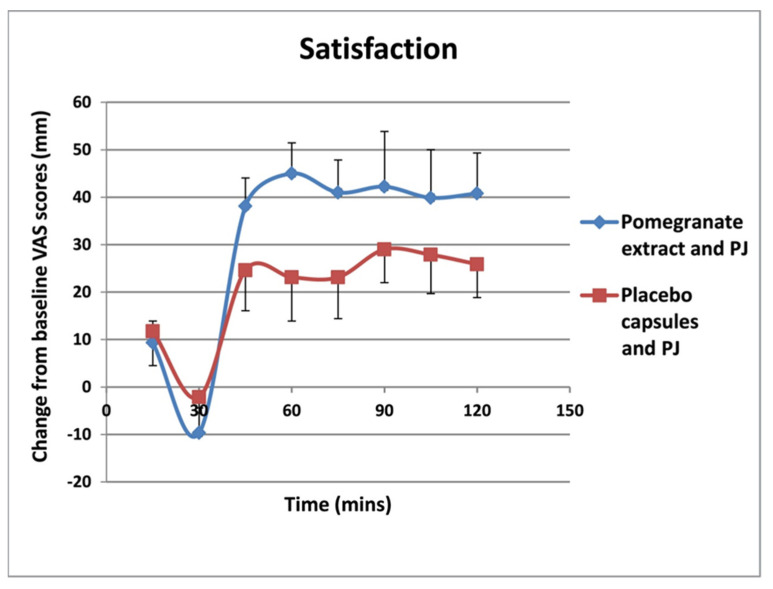
Change in hunger from baseline (mm VAS) over 120-min lunch meal period after 3 weeks of PE or placebo capsule consumption with pre-meal PJ intake. Results are expressed as mean (±SD). PJ: pomegranate juice. There was an overall statistically significant difference between the two groups (*p* = 0.036) with subjects in the PE group being significantly more satisfied.

**Figure 4 foods-11-02639-f004:**
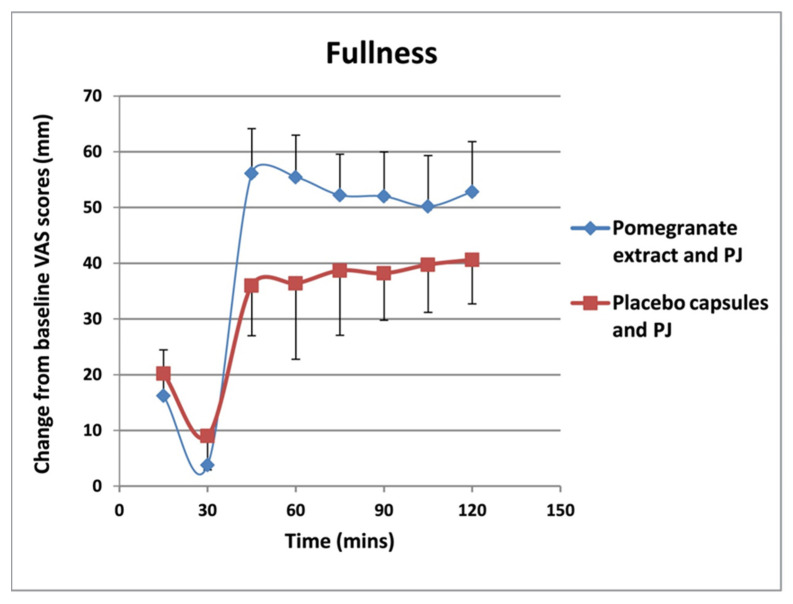
Change in fullness scores from baseline (mm VAS) over 120-min lunch meal period after 3 weeks of PE or placebo capsule consumption with pre-meal PJ intake. Results are expressed as mean (±SD). PJ: pomegranate juice. Subjects in the PE group experienced a greater feeling of fullness than those in the placebo group with PJ preload, which was statistically significant (*p* = 0.02).

**Figure 5 foods-11-02639-f005:**
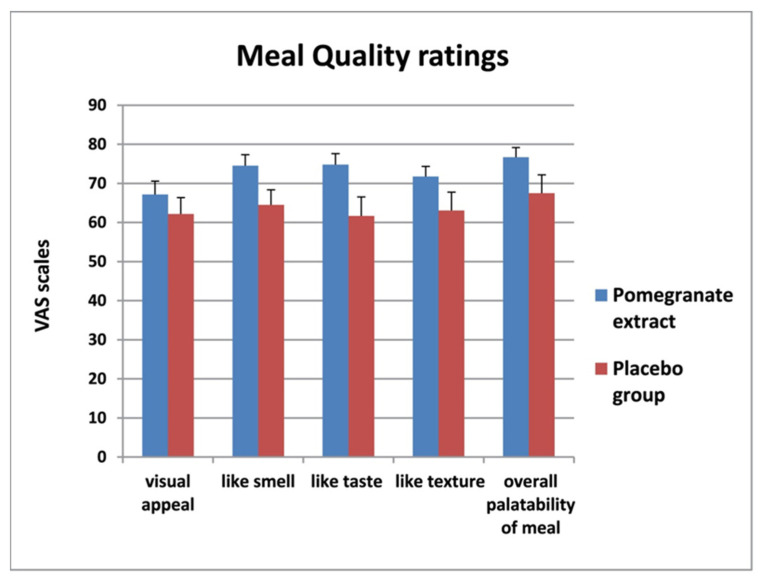
Meal quality ratings for the pomegranate (PE) and placebo priming groups. All VAS scores for individual characteristics were higher in the PE group as compared with placebo. The PE group liked the smell of the meal significantly more than the placebo group (*p* = 0.02) (Data as mean ± SD).

**Figure 6 foods-11-02639-f006:**
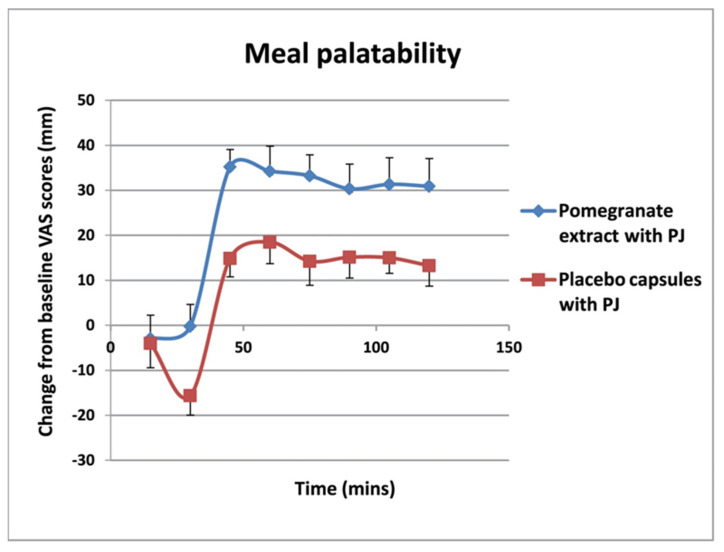
Meal palatability ratings over the 120-min lunch period for the pomegranate extract (PE) and placebo capsules (PL) group with pomegranate juice (PJ) preload. The meal palatability was rated significantly more highly (*p* = 0.05) by the PE group with the PJ preload compared to those in the placebo group (data as mean ± SD).

**Table 1 foods-11-02639-t001:** Nutritional composition of the lunch meal and drinks.

Nutritional Composition	Pasta (500 g Cooked)	Pasta Sauce (500 g)	Total Meal Composition	Pomegranate Juice (PJ)	Placebo Juice
Energy (kcal)	765	220	985	72	72
Protein (g)	28	9	37	0.6	Trace
Carbohydrates (g)	138.5	39.5	178	18	18
Sugars (g)	3.5	35	38.5		
Fat (g)	8.5	3	11.5	Trace	Trace
Saturated fat (g)	2.5	0.5	3		
Fibre (g)	12	9	21	Trace	Trace
Sodium (g)	Trace	2	2		
Polyphenols (mg GAE)				126	Negligible

GAE: gallic acid equivalent.

**Table 2 foods-11-02639-t002:** Satiety study intervention protocol.

Time	Procedures
8:30 a.m.	Breakfast served and eaten. 60 g crunchy nut cornflakes; 150 mL semi-skim milk; 150 mL pomegranate juice (PJ) or placebo drink (PLD).
3-h interval	Participants fast and then returned to the kitchen/food laboratory for the lunch session.
11:30 a.m. (0–15 min)	Starting to complete VAS for satiety (VAS 1 baseline; zero point) 150 mL of PJ juice preload for all participants (both groups) (VAS 2 satiety and palatability variables).
12:00 noon (30 min)	Lunch presented to each participant (VAS 3 satiety variables before eating anything)
12:05 p.m. (35 min)	Participants consumed 1 spoon of lunch (VAS for palatability and meal quality variables). Afterwards, participants continued to eat the lunch.
12:15 p.m. (45 min)	Participants completed VAS 4 for satiety and meal palatability variables. A timer prompts the participants.
12:30 p.m. (60 min)	VAS 5; satiety and meal palatability (repeat VAS 4)
12:45 p.m. (75 min)	VAS 6; satiety and meal palatability (repeat VAS 4)
1:00 p.m. (90 min)	VAS 7; satiety and meal palatability (repeat VAS 4)
1:15 p.m. (105 min)	VAS 8; satiety and meal palatability (repeat VAS 4)
1:30 p.m. (120 min)	VAS 9; satiety and meal palatability (repeat VAS 4)

**Table 3 foods-11-02639-t003:** Baseline characteristics of subjects randomized to the study groups.

Characteristic	Pomegranate Extract Capsule Group Mean (SD)	Placebo Capsule Group Mean (SD)
Age (years)	34.5 (13.7)	32.6 (12.9)
Gender (M; F)	4; 10	4; 10
Height (m)	1.66 (0.06)	1.70 (0.09)
Weight (kg)	70.0 (13.7)	72.1 (12.0)
BMI (kg/m^2^)	25.1 (4.5)	24.94 (3.4)

**Table 4 foods-11-02639-t004:** Satiety study energy expenditure and macronutrient intake at baseline and 3-week post study values for PE and PL groups with differences (baseline—3 weeks). Results are expressed in means with SD and SEM and *p* values. There were significant increases in carbohydrates, sugars, and starch in both groups.

Variable	Pomegranate Extract (PE)							Placebo (PL) Group (*n* = 14)	
	Group (*n* = 14)								
					*p*				*p*
		Mean	SD	SEM	Value	Mean	SD	SEM	Value
Energy (kcal)	Baseline	1913.20	415.97	107.40		1840.64	411.93	113.19	
	4th week	2109.53	512.60	132.35		2119.79	542.34	143.30	
	Change	−196.33	388.90	100.41	0.071	−279.14	629.49	168.24	0.121
Energy (kJ)	Baseline	8037.53	1744.32	450.38		7738.14	322.55	86.21	
	4th week	8893.00	2148.51	554.74		8931.14	556.36	148.69	
	Change	−855.47	1611.12	415.99	0.069	1193.00	2639.54	705.45	0.115
Protein (g)	Baseline	75.52	16.42	4.24		78.44	1345.88	359.70	
	4th week	75.23	12.69	3.28		91.34	2338.91	625.10	
	Change	0.29	17.48	4.51	0.950	−12.89	24.31	6.50	0.069
Carbohydrate (g)	Baseline	232.75	59.84	15.45		210.76	38.45	10.28	
	4th week	297.53	74.52	19.24		269.29	54.24	14.50	
	Change	−64.78	30.69	7.92	0.021	−58.53	80.59	21.54	0.018
Sugars (g)	Baseline	101.47	42.59	11.00		90.91	34.67	9.27	
	4th week	129.14	47.02	12.14		111.00	67.11	17.94	
	Change	−27.67	30.17	7.79	0.033	−20.09	29.12	7.78	0.023
Starch (g)	Baseline	127.12	44.23	11.42		112.13	27.58	7.37	
	4th week	162.75	44.19	11.41		154.53	26.64	7.12	
	Change	−35.63	31.52	8.14	0.021	−42.40	60.00	16.04	0.026
Total fat (g)	Baseline	76.73	30.22	7.80		80.28	30.08	8.04	
	4th week	71.60	26.46	6.83		77.90	44.82	11.98	
	Change	5.13	32.90	8.50	0.556	2.38	33.40	8.93	0.794

PE: pomegranate extract; PL: placebo capsules; SD: standard deviation; SEM: standard error of the mean.

## Data Availability

Study data are available from the authors upon request.

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
