# Peer review of "Effect of Pomegranate Extract Consumption on Satiety Parameters in Healthy Volunteers: A Preliminary Randomized Study"

_foods, 2022, doi:10.3390/foods11172639_

Round 1
Reviewer 1 Report
The MS investigated the potential application of PJ in modulating satiety. The data is plenty, and results are interesting. The authors may need to clarify the following concerns.
· Table 1: No sugars in PJ and placebo. What are the major components of carbohydrates in PJ? Which contributes to energy?
· Why PJ is served for both groups? Will it create some noise on data analysis?
· Fig 1: why the first data of Placebo is lower than PE? Similarly, data at 30 min in Fig 3&4 is also abnormal.
· The MS showed that “lower levels of hunger and a desire to eat, as well as higher levels of 319 fullness and satisfaction, thus greater levels of satiety, in participants consuming PE with 320 PJ, compared to placebo.” However, participants of PE group gave higher ranking for meal quality and palatability, which seems contradictory.
Author Response
Please see a point-by-point response to the reviewer’s comments in the file attached.

Reviewer 2 Report
The present study focused on the effect of oral pomegranate extract and pomegranate extract juice on satiety parameters in healthy volunteers, and found that consumption of pomegranate extract could have the potential to modulate satiety indicators. Although the authors found differences between groups, it is undeniable that these indicators have subjective tendencies which were different from objective detection indicators. Different people may have different evaluation standards for those indicators. In addition, although the authors found that appetite related indicators decreased, the intake of macronutrients increased. The author needs to give a reasonable explanation.
1) Line 53-71, is there any animal experiment that shows that polyphenols can affect appetite. The author should describe the effect of polyphenols on appetite and its potential mechanism in the introduction section.
2) How to distinguish whether the change of appetite is the effect of polyphenol extraction intake for three weeks or the effect of pomegranate extract juice on the day of test?
3) Whether the control group also drank pomegranate extract juice on the same day was not consistent with the whole manuscript. For example, authors stated that “the placebo juice was composed of diluted orange juice with very low polyphenols containing the same amount of energy as the PJ by adding sucrose” (line 119-12-), which mean the placebo did not intake pomegranate extract. But the figure 1-6, placebo capsules with PJ mean the participants in the placebo group intake the PJ?
4) 4.1: The effects of extracts on glucose and lipid metabolism and obesity were discussed, but not on appetite. The author should discuss the underlying mechanism by which polyphenols affect appetite.
Author Response
Please see a point-by-point response in the attached file.

Reviewer 3 Report
Thanks for the opportunity to review this research. The manuscript entitled „Effect of pomegranate extract consumption on satiety parameters in healthy volunteers: a preliminary randomized study” have described the the effect of oral pomegranate extract and juice intake vs. placebo on satiety parameters in healthy volunteers.
1. There are several typographical mistakes as well in whole manuscript. Therefore, the author’s thoroughly careful check the language and typo mistake to minimize the error.
2. The abstract should be beginning with a sentence about the background of concept and the aims as well as novelty of study should be mentions. What exactly is the novelty of this study? Abbreviations must be avoided in abstract. Parenthesis should be avoided in abstract. Please improve.
3. Introduction; Check and format the citations in the whole manuscript. Also, Appropriate references must be provided to explained the background, what is already done and why this study carried out. Other vise the novelty of this research is still poorly presented. This is important especially for the high IF journals. The scientific style should be used. Hypothesis statement is missing in the introduction section.
4. Material and methods; The whole M&M section must be substantially rewritten and improved. The methods are not properly referenced and are not possible to follow, reproduce and verify.
5. Results and discussion; General remark to the discussion - In my opinion, the discussion provided by Authors is difficult to follow and verify due missing critical details in the methodology section.
Author Response

(The authors gave the same response as above.)

Round 2
Reviewer 1 Report
The response has addressed some problems and can be more comprehensive
Reviewer 2 Report
All comments received positive responses. I have no more suggestions.